# A Method for High-Value Driving Demonstration Data Generation Based on One-Dimensional Deep Convolutional Generative Adversarial Networks

Yukun Wu , Xuncheng Wu *, Siyuan Qiu and Wenbin Xiang

School of Mechanical & Automotive Engineering, Shanghai University of Engineering Science, Shanghai 201620, China
* Correspondence: m010120451@sues.edu.cn

**Abstract:** As a promising sequential decision-making algorithm, deep reinforcement learning (RL) has been applied in many fields. However, the related methods often demand a large amount of time before they can achieve acceptable performance. While learning from demonstration has greatly improved reinforcement learning efficiency, it poses some challenges. In the past, it has required collecting demonstration data from controllers (either human or controller). However, demonstration data are not always available in some sparse reward tasks. Most importantly, there exist unknown differences between agents and human experts in observing the environment. This means that not all of the human expert's demonstration data conform to a Markov decision process (MDP). In this paper, a method of reinforcement learning from generated data (RLfGD) is presented, and consists of a generative model and a learning model. The generative model introduces a method to generate the demonstration data with a one-dimensional deep convolutional generative adversarial network. The learning model applies the demonstration data to the reinforcement learning process to greatly improve the effectiveness of training. Two complex traffic scenarios were tested to evaluate the proposed algorithm. The experimental results demonstrate that RLfGD is capable of obtaining higher scores more quickly than DDQN in both of two complex traffic scenarios. The performance of reinforcement learning algorithms can be greatly improved with this approach to sparse reward problems.

**Keywords:** deep Q-networks; deep convolutional generative adversarial network; autonomous vehicle; decision making; experience replay

## 1. Introduction

Reinforcement learning is one of the most popular artificial intelligence fields, in which the agent learns a policy through interaction with the environment. The introduction of fixed Q-targets and experience replay in deep Q-networks (DQN) [1] has greatly contributed to the development of reinforcement learning. In the past several years, these techniques have been successful with sequential decision tasks such as robot control [2], natural language processing (NLP) [3], autonomous vehicle decision-making [4,5], etc. However, most RL algorithms are still in their infancy and have very limited real-world applications. One of the greatest challenges for RL is the difficulty in achieving convergence. Developing a reasonable policy requires a great deal of trial and error in the environment, and the reward function needs to be well designed. Imitation learning (IL) [6] differs from reinforcement learning since it does not develop optimal policy through accumulating rewards. In IL, the agent learns policy from human demonstration. Three main approaches are available: behavioral cloning [7], inverse reinforcement learning [8], and generative adversarial imitation learning (GAIL) [9]. IL is more effective than RL in sparse reward tasks thanks to the human demonstration. However, there are some disadvantages associated with IL. Typically, it requires a large amount of expert demonstration data, and collecting

expert demonstration data can be expensive. Additionally, human presentation data may not always follow MDP [10], and the actions taken by experts may not entirely depend on what they observe at the moment. In addition, imitation learning [11] is also limited by error accumulation and multi-modal problems.

Combining reinforcement learning and imitation learning is more consistent with the human learning process, which is shown to help in sparse reward problems. Combining RL and IL research has been conducted for policy shaping [12], reward shaping [13], and knowledge transfer with human demonstration [14]. Demonstration data [15], which prove useful for sparse reward tasks, provide a great bridge between the two. Experience replay (ER) [16] is one of the key techniques for DQN to reach the human level. Putting the demonstration data into the RL's experience replay is a very creative way to combine RL and IL. This method is named reinforcement learning with expert demonstrations (RLED) [17,18]. Following this framework, such algorithms are presented. In human experience replay [19], the agent sample from the replay buffer mixes the agent's transitions and demonstration data during training. Replay buffer spike is another algorithm that uses a demonstration data initial replay buffer. However, both of these do not pre-train the agent or keep the demonstration data in a replay buffer. Accelerated DQN with expert trajectories [20] uses a combination of TD and cross-entropy losses in the DQN learning process, while not pre-training the agent for good initial performance. Deep Q-learning from the demonstration (DQfD) [21] is an advanced framework of learning from demonstration. It introduces a supervised loss and an L2 regularization loss [22] to pre-train the agent and update the target network. Experiments show that DQfD is superior to double DQN and IL for 27 and 31 of 42 Atari games, respectively. While the algorithm achieves excellent results, collecting data for the demonstration of some complex problems is challenging, which limits its application.

Demonstration data consist of many transitions. Not all transitions are useful for training; training procession prefers benefits from high-reward transitions [23]. Nevertheless, high-reward transitions are not always available, especially in some sparse reward tasks. In addition, the human expert demonstration may not always follow MDP, since the differences between humans and agents when observing the environment are particularly evident for complex tasks. In particular, when it comes to autonomous vehicle decision making, not all human expert demonstrations are available. Human drivers often possess a wide range of driving experience and do not always make decisions based on current observations. Considering the difficulty of collecting demonstration data on sparse reward problems and the different ways of observing the environment between humans and agents, in this paper, a method of reinforcement learning from generated data (RLfGD) is presented, which consists of a generative model and a learning model. The generative model, named one-dimensional deep convolutional generative adversarial network (1-DGAN), which is built on top of deep convolutional generative adversarial networks (DCGANs) [24], introduces a method to generate high-value demonstration data. To generate one-dimensional data, both the generator and the discriminator are built from one-dimensional convolutional neural networks. In addition, classification networks are trained to address the inability to generate discrete action information in demonstration data. The learning model based on DQfD places the high-value demonstration data in the DQN experience buffer and samples them using a high-value priority replay method, greatly enhancing the efficiency of the DQN. The proposed method is expected to provide a large amount of reasonable driving demonstration data to improve the performance of reinforcement learning algorithms in the autonomous vehicle field.

The paper is structured as follows: The literature review is shown in Section 2. The model is built in Section 3. The experimental setup and experiments' results are described in Section 4, and the conclusion and future works are shown in Section 5.

## 2. Literature Review

As demonstration is shown to help in sparse reward problems [25], an increasing amount of the literature has been concerned with learning from demonstration. Jessica et al. introduced DPID algorithms [26], which naturally combine RL and demonstration data. A theoretical analysis of how RL problems can be solved using demonstration data was provided in this work. In [27], Mueller investigated the integration of high-level abstract behavioral requirements into learning from demonstration (LfD) for robots. This method enforces high-level constrains on a part of the robot's motion plan during training. A major constrain of this work is that it needs to encode motion planning constraints. Anahita et al. [28] applied the demonstration data to trajectory learning for robots. However, this method needs teleoperation to collect the demonstration data and cannot be applied to autonomous vehicles. Ashvin et al. [29] utilized the demonstration data to overcome agent exploration in RL. In [30], a model based on inverse reinforcement learning that learns driving preferences from demonstration data was proposed. Christian et al. [31] used human data to pre-train the policy network at first and later tuned by RL, and placed third in the NeurIPS MineRL Competition. In [32], Yichuan et al. developed a novel LfD method that uses the Bayesian network to extract human knowledge from demonstrations. These related studies do not address the problem of inconvenient demonstration data collection.

In [33], Mel et al. combined the demonstration data and the deep deterministic policy gradients (DDPGs) algorithm to propose DDPG with demonstration. Demonstrations were collected by robots controlled by humans. This method uses the demonstration to replace elaborately engineered rewards, and outperformed DDPG on real robotics tasks controlling the movement of the robot arm. However, this work does not pre-train the agent using the demonstration data, so the agent cannot update with TD updates when the agent initially interacts with the environment. Sixiang et al. [34] proposed another DDPG with demonstration algorithm following the DQfD framework. This algorithm uses the combined loss function to make the agent learn human demonstration policy, and the experience replay buffer is also populated from various transition data samples. The results of the experiment show that it improves training efficiency and the potential in mastering human preference. Kai et al. [35] also combined DDPG with expert demonstration; the difference with previous work is that they integrated reward construction with the training process. Similarly, Lei et al. [36] introduced the demonstration data to the twin delayed DDPG (TD3) algorithm and achieved success in a challenging 3D UAV navigation problem. All of these researchers used human expert demonstration and did not address the difficulties of data collection. Evan et al. [37] developed a method based on SAC and hindsight experience replay (HER), which provides an order of magnitude of speedup over RL on simulated robotics tasks. However, this method requires several demonstration data, which are inefficient to sample under complex tasks. Abhik et al. [38] proposed a UAV obstacle avoidance method that uses GAN architecture and the DQfD framework. This work is the one most closely related to us, but it used GAN networks to generate depth images from RGB images rather than generating demonstration data. Both of these related works utilized demonstration data to enhance the performance of RL algorithms. The introduction of human demonstration data makes it possible to limit the exploration interval of agents to a reasonable range. However, the collection of demonstrations is still a hindrance in many complex sparse reward problems. Research on how to collection demonstration data in sparse problems has yet to be developed.

## 3. Materials and Methods

In this section, a learning model and a generative model are built. The generative model named 1-DGAN was built on top of the DCGAN framework [39,40]. The learning model named RLfGD was based on the DQfD framework.

*3.1. Built the Generative Model*

GAN [41] consists of two networks: a generator *G* that generates data similar to real data and a discriminator *D* that determines whether data are generated or real. *D* is trained to attribute the correct label to training samples. Simultaneously, *G* is trained to generate the same data as the real data to be able to cheat the discriminator. *G* and *D* constitute a dynamic game process in which the capabilities of both parties continue to improve through iteration. The objective function of the GAN model is:

$$\min_G \max_D V(D,G) = E_{x \sim p_{\mathrm{data}}(x)}[\log D(x)] + E_{z \sim p_z(z)}[\log(1 - D(G(z)))] \tag{1}$$

where *D* is a discriminative network with real data *x* as input and *G* is a generative network with random noise *z* as input; $D(x)$ indicates the probability that the input *x* are an actual sample; and $D(G(z))$ denotes the probability that the generated data d are a true sample. The loss function of Formula (6) is decomposed into two parts: discriminative model loss function and generative model loss function. *D* is trained by the discriminative model loss function for correctly assigning the label to the input data (real data: 1, generated data: 0); *G* is simultaneously trained by the generative model loss function to minimize $\log(1 - D(G(z)))$.

The proposed deep convolutional GAN (DAGAN) [24] solves the problem of unstable training of typical GAN. Both the generator and discriminator of DCGAN discard the pooling layer of CNN [42], the discriminator keeps the overall architecture of CNN, and the generator replaces the convolutional layer with fractional-strided convolution [43]. The batch normalization (BN) [44] layer is used after each layer in the discriminator and generator and helps deal with training problems caused by poor initialization as well as accelerating model training and improving the stability of training. In addition, all layers use the ReLu activation function except for the output layer, which uses the Tanh activation function. The LeakyReLU [45] activation function is used in all layers of the discriminator to prevent gradient sparsity.

DCGAN [24] greatly improves the stability of the original GAN training and the quality of the generated results. The digital images are regarded as continuous data for GAN since the pixel values vary continuously in the range of 0–255. There has been a great success with GAN in the area of image generation. With discrete data, the discriminative network cannot backpropagate the gradient to the generative network. Therefore, it performs poorly when it comes to text generation.

The typical DCGAN with a 2D convolutional network as the backbone is used to generate images. However, in this paper the driving demonstrations are defined as 1D data. Consequently, the proposed generative model is based on the 1D convolutional network for process sequence data.

As shown in Figure 1, two networks are built: a generator and a discriminator. The generator *G* generates fake data from input random noise **z** that is usually uniform noise while the discriminator D inputs both fake data $G(z)$ and true data and outputs the probability $D(x)$ that the data are true. In *G* and *D*, both utilize the "Conv1D + BN + LeakRuLe" structure. Both the generator and discriminator use one-dimensional convolutional neural networks, since they are more suited to producing one-dimensional demonstration data. For *G*, the first layer is the full connection layer, and the activation function of the output layer is tanh. Both the discriminator and the generator contain a fully connected neural network, which increases the network's complexity and prevents overfitting. In addition, the real data are normalized before being fed into the discriminator.

The main process of 1-DGAN is as follows: First, the generation network generates demonstration data randomly. The quality of the generated data are poor at the beginning, and the discriminator can easily distinguish the generated data from the real data. Next, the generator is trained to generate data that can deceive the untrained discriminator. Then, the discriminator is trained to discriminate against the data generated by the trained generator as

false. The generator and discriminator are continuously trained according to Equation (1) until the discriminator is unable to distinguish between real and generated data.

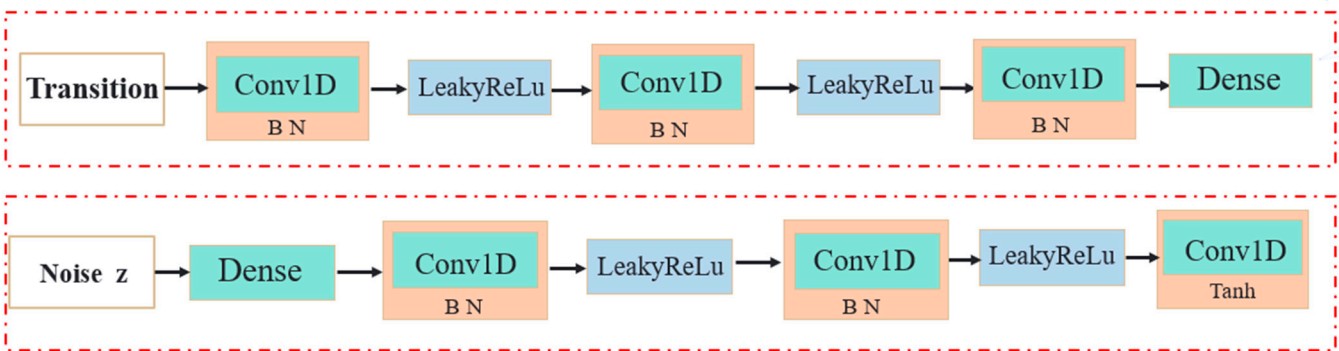

**Figure 1.** The architecture of the generator (down) and discriminator (up).

*3.2. Built Learning Model*

An MDP framework is used in this study, and can be described as a tuple $<S, A, T, R, \gamma>$, where $S$ is the state space; $A$ is the action space; $T$ is the state transition function $T(s, s', a) = P(s'|s, a)$; and $\gamma$ is the discount factor. The agent interacts with the environment following the MDP. In each step $t$, the agent takes an action $a_t$, making the environment state transfer from $S_t$ to $S_{t+1}$, and a reward $R_t$ is received, indicating the performance of the transition. In $Q$-learning, the aim is to maximize the $Q$ value function according to the Bellman equation with the following structure:

$$Q(s_t, a_t) = R_t + \gamma \max_{a_{t+1}} Q(s_{t+1}, a_{t+1}) \tag{2}$$

DQN [1] not only leverages a deep convolutional neural network to approximate the $Q$-function but also introduces experience replay to store and reuse transitions' sequences and target networks to address the overestimation problem. The DQN algorithm exploits the transitions and randomly samples from the replay buffer to update the network using the minimizing loss function:

$$L(\theta_t) = \sum_1^N \left( R_t + \gamma \max_a Q(s_{t+1}, a; \theta_t^-) - Q(s_t, a_t; \theta_t) \right)^2 \tag{3}$$

where $\theta_t^-$ is the target network parameter, $\theta_t$ is the $Q$-network parameter, and $\theta_t^-$ is updated by $\theta_t$ only every k time step. The loss function trains $\theta_t$:

$$\nabla_{\theta_t} L(\theta_t) = \left( r_t + \gamma \max_a Q(s_{t+1}, a; \theta_t^-) - Q(s_t, a_t; \theta_t) \right) \nabla_{\theta_r} Q(s_t, a_t; \theta_t) \tag{4}$$

Double deep $Q$-network (DDQN) [46], an extension of the nature DQN, eliminates the overestimation problem by decoupling the steps of selecting the target $Q$-action and calculating the target $Q$-value. In DDQN, two value functions are available, with parameters of $\theta$ and $\theta'$:

$$Y_t^{\text{DQN}} \equiv R_{t+1} + \gamma \max_a Q(S_{t+1}, a; \theta_t^-) \tag{5}$$

$$Y_t^{\text{DoubleQ}} \equiv R_{t+1} + \gamma Q(S_{t+1}, \text{argmax} Q(S_{t+1}, a; \theta_t); \theta_t') \tag{6}$$

where $Yt^{DQN}$ is the DQN target value and $Yt^{DoubleQ}$ is the DDQN target value. It still uses a greedy policy to estimate the $Q$ value based on the current value defined by $\theta$. However, the second set of weight parameters $\theta'$ is used to evaluate this policy value.

In this study, an algorithm RLfGD is presented based on the awesome work of DQfD [21], the framework shown in Figure 2. In order to reduce the exploration space of the agent, it expects to match the behavior of the demonstrator as closely as possible before interacting with the environment. In RLfGD, the demonstration data are used for the pre-training agent, and the agent updates the network by applying four losses: the

"1-step" and "n-step" double $Q$-learning losses, a large margin classification loss, and an L2 regularization loss. The agent is to imitate the previous controller with a value function that fulfills the Bellman equation:

$$J_{DQ}(Q) = \left[ R(s,a) + \gamma Q\left(s_{t+1}, a_{t+1}^{\max}; \theta^-\right) \right] - Q(s_t, a_t; \theta)) \tag{7}$$

$$J_E(Q) = \max_{a \in A}[Q(s,a) + l(s,a_E,a)] - Q(s,a_E) \tag{8}$$

$$J(Q) = J_{DQ} + \lambda_1 J_E(Q) + \lambda_2 J_{L2}(Q) + \lambda_3 J_n(Q) \tag{9}$$

where $J_{DQ}$ is the loss of DDQN, $J_E$ supervises a large margin loss, $J_{L2}$ is $L2$ regularization losses, and $\lambda_1$, $\lambda_2$, and $\lambda_3$ are parameters that adjust the weights between losses.

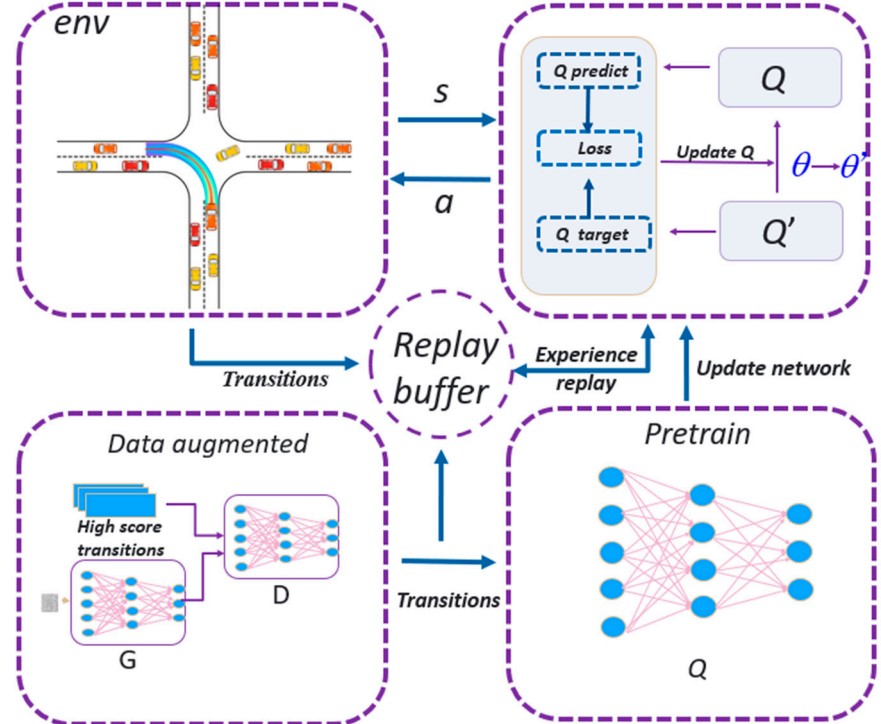

**Figure 2.** The framework of RLfGD.

After pre-training, agents begin to interact with the environment following the MDP. At the end of a step, a transition will be received: a slightly high-reward transition will be stored in the replay buffer as experience. The requirements for rewards from transition generation in the training phrase are lower than the requirements for the demonstration data. In particular, this method keeps the demonstration data in the replay buffer permanently, but the experience generated during training will be replaced by a new experience when the replay buffer is full. This means that the demonstration data are never rewritten. During the training phase, the agent samples mini-batch n transitions from demonstration according to high-value prioritized experience replay (HVPER) [23]. In HVPER, the prioritizing of transitions is calculated by the reward, state–action value, and TD error. Compared to priority experience replay [47], HVPER accelerates the training process and reaches a high performance.

The process of RLfGD is described as follows: The first step is to record the demonstration data from the previous system (non-human experts), and only concentrate on transitions with a high reward. Secondly, generate sufficient demonstration data based on a small set of pre-processed high-reward transitions using 1-DGAN. Then the demonstration data are used for the pre-training agent, and the agent updates the network by applying four losses. Finally, the agent updates its strategy during reinforcement learning in order to reach convergence.

## 4. Experiments

To test the validity of reinforcement learning from generative data (RLfGD), experiments were performed in the highway_env [48] environment, which contains 6 traffic tasks such as highways, intersections, etc. We considered the DDQN algorithm as the baseline, and compared the RLfGD with DDQN on two complex traffic scenarios: highway overtaking and intersection turn-left. Every comparison experiment was conducted using the same neural network architecture, randomized seed, and evaluation setup.

### 4.1. Data Process

The demonstration data are composed of transitions generated at each time step. These transitions can be described as a tuple (state, action, reward, next_state). The dimension size of each element is described in Table 1. The state can be described as the position and speed of the vehicles; the specific description is shown in Table 2. It continuously changes values within a certain range, with high fault tolerance, and are continuous data for GAN. Similarly, the reward is continuous data. However, the action is several discrete numbers, and different numbers represent different actions. Therefore, action information is discrete data for GAN. With discrete data, the discriminative network cannot backpropagate the gradient to the generative network. This means that discrete action information cannot be generated using 1-DGAN.

**Table 1.** The dimension of state space.

|            | Highway       | Intersection   |
|------------|---------------|----------------|
| State      | $5 \times 5$  | $15 \times 7$  |
| Action     | $3 \times 1$  | $5 \times 1$   |
| Reward     | $1 \times 1$  | $1 \times 1$   |
| next_state | $5 \times 5$  | $15 \times 7$  |

**Table 2.** The description of state space parameters.

| Feature   | Description                                                                                           |
|-----------|------------------------------------------------------------------------------------------------------|
| $P$       | Disambiguate agents at 0 offsets from non-existent agents                                             |
| $x$       | The offset of the world vehicle as it relates to the ego vehicle on the x-axis or the offset to the ego vehicle |
| $y$       | The offset of the world vehicle as it relates to the ego vehicle on the y-axis or the offset to the ego vehicle |
| $v_x$     | The speed of a vehicle on the x-axis                                                                  |
| $v_y$     | The speed of a vehicle on the x-axis                                                                  |
| $cos_h$   | The triangular heading of vehicles                                                                    |
| $sin_h$   | The triangular heading of vehicles                                                                    |

To address the above problem, a classification network was developed using states as data and actions as labels. In RL, the agent selects an action based on the current state, and the environment updates the state according to the state transition function after the agent makes the action. This means that there is a mapping between state and action. Consequently, the problem can be considered a classification issue, and a neural network can be trained to discover this mapping relationship.

The neural network can predict the corresponding state action per neural network, so the 1-DGAN is only used to generate state information and avoid the input of discreet action information. The training curve is shown in Figure 3; both of them exhibit strong performance on the test and validation sets. In the intersection environment, the accuracies of the model on the test machine and validation set are 83.32% and 78.57%. In the highway environment, the accuracies of the model are 86.23% and 81.68%.

After the training, the classification network can predict actions based on state. In particular, a parameter in the state named the present of ego-vehicle (description in Table 2)

is always described as "1", so the normalized reward can be used to replace that. This is particularly significant for the processing of two-dimensional input features; if the neural network takes the two-dimensional state information as input, the reward information is treated as an additional feature. However, this paper reshapes the state information into a one-dimensional vector, so the normalized reward information can be directly added to the vector. Afterward, these features are fed into the discriminator. Discriminators and generators game with each other to generate fake demo data.

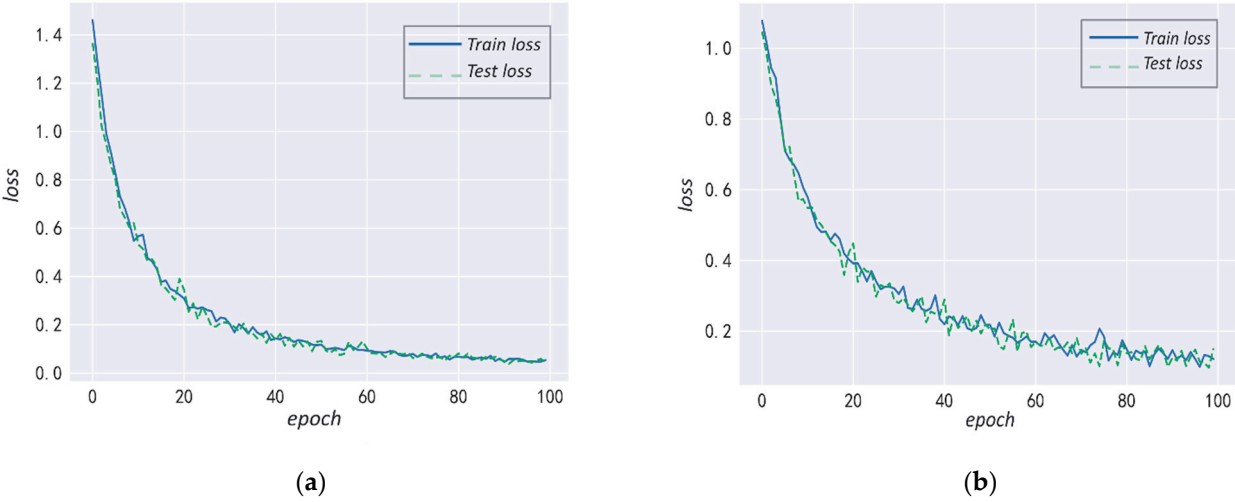

(**a**)                                                    (**b**)

**Figure 3.** The loss curve of the state–action classification network, picture (**a**) is the loss curve of the highway environment; (**b**) is the loss curve of the intersection environment.

*4.2. Highway Environment*

As shown in Figure 4, in the highway task, an ego-vehicle is driving on a multi-lane highway road populated with other vehicles. The vehicle needs to change lanes to avoid colliding with another vehicle. The vehicle's objective is to reach a high speed as quickly as possible while remaining in the rightmost lane.

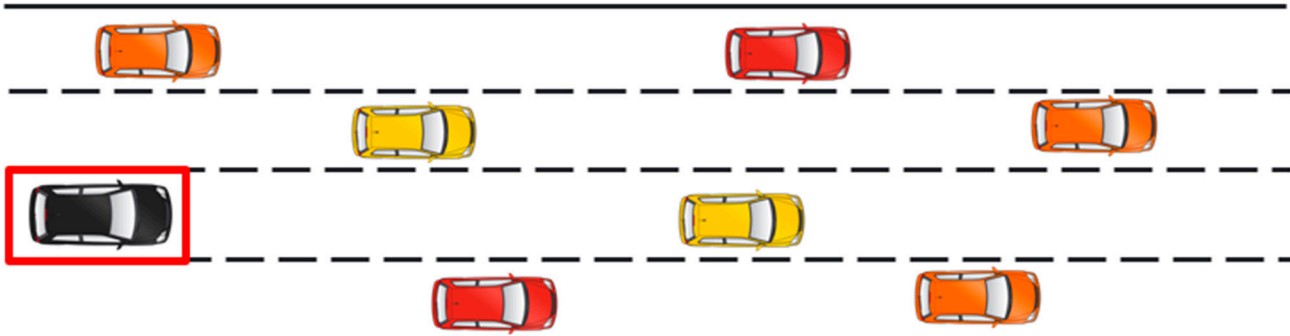

**Figure 4.** The highway environment (vehicle in the red box is ego-vehicle).

The state space S is described by the $V \times F$ array that describes a list of $V$ nearby vehicles by a set of features of size $F$, where S = $[P, x, y, v_x, v_y]$. The agent observes the five vehicles that are closest to it, and the size of the state is a $5 \times 5$ array.

The action space consists of five movement types: acceleration, deceleration, left lane change, right lane change, and idle. For the sake of reasonableness, several actions are not permitted in some states: changing lanes at the edge of the road, or accelerating/decelerating over the maximum/minimum speed. The agent is rewarded by reaching a high velocity or remaining in the rightmost lane and avoiding collisions. Otherwise, a negative reward will be given.

$$R = \begin{cases} -1, \text{ when colliding with a vehicle} \\ 0.6, \text{ when driving at full speed} \\ 0.2, \text{ when driving on the right} - \text{most lanes} \\ 0, \text{ at each lane change action} \end{cases} \tag{10}$$

The ego-vehicle is driven by the agent, and other vehicles in the environment are controlled by the IDMVehicle dynamics model. The experiment performed 1000 episodes of training. The end of every epoch was when a collision occurred or the time exceeded 40 s.

*4.3. Intersection Environment*

Traveling at unsignalized intersections is one of the most complex and dangerous traffic scenarios for autonomous vehicles. Due to the complexity of unsignalized intersections, this is still a challenging task. We present an intersection negotiation task with dense traffic composed of two parallel roads and several traffic participants, as shown in Figure 5. The agent drives the ego-vehicle from south to west; during traveling, the agent interacts with the traffic participants controlled by the IDMVehicle dynamics model, aims to avoid collision, and tries to leave the intersection at the desired speed. The traffic participants' position and destination are randomly initialized.

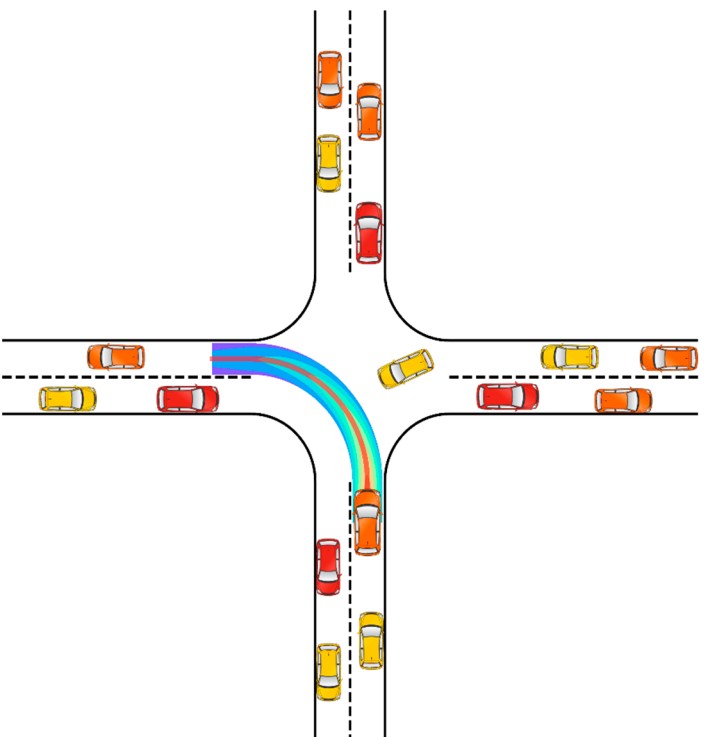

**Figure 5.** The highway environment.

The state space $S$ is also described by the $V \times F$ array, $S = [P, x, y, vx, vy, cosh, sinh]$. The agent observes the seven vehicles that are closest to it; the size of the state is a $15 \times 7$ array. Low-level controllers implement lateral control for vehicles automatically. The agent only controls the acceleration of its vehicle. Therefore, the action space consists of three movement types: acceleration, deceleration, and idle. In general, rewards are defined as follows:

$$R = \begin{cases} -5 \text{ when a collision occurs} \\ 1 \text{ when it drives at maximum velocity} \\ 0 \text{ otherwise} \end{cases} \tag{11}$$

This experiment performed 3000 episodes of training. Epochs ended when collisions occurred or the step time exceeded 13 s.

### 4.4. Experimental Results

Firstly, to evaluate the validity of the generative module, we visualized the distribution of generated data and real data. The histogram of two types of data, real and partially generated, is shown in Figure 6 for two environments; the demonstration data generated by the generative model have the same distribution as the real demo data. In Figure 6, Figure 6a,b shown that the generated data also have a high longitudinal velocity. The policy of lateral control of the generated data is also largely coherent with the actual data, as shown in Figure 6c,d.

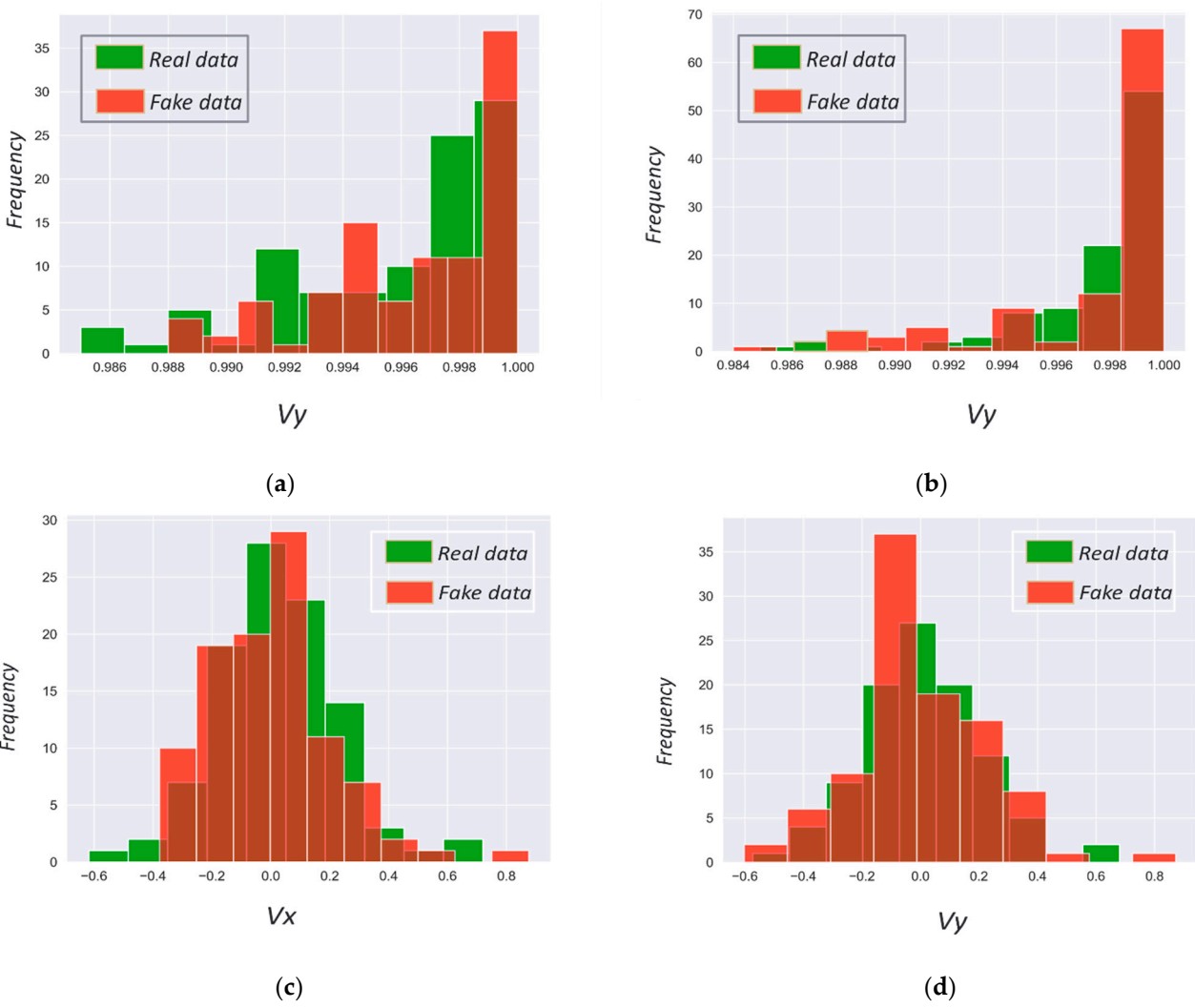

**Figure 6.** In the histogram of partially generated data and real data, Figures (**a**,**b**) are the longitudinal speed of the vehicle at the intersection (**a**) and highway (**b**) environment; Figures (**c**,**d**) are the lateral speed of the vehicle at the intersection (**c**) and highway (**d**) environment. The vehicle's lateral and longitudinal speeds are normalized.

In addition, the real and generated trajectories of ego-vehicles in intersection environments are also visualized in Figure 7, in which the generating trajectory is approximately the same as the real trajectory. As a result of visualizing both types of data, it can be demonstrated that the proposed generative model can generate data that are very close to the **real** data, although there are some errors. However, the vehicle coordinate data have a high tolerance rate that can accept minute errors.

To further verify the validity of the generated demo data, RLfGD was deployed in two complex traffic scenarios. The learning curve is shown in Figure 8 for two traffic scenarios: highway and intersection. On the highway, the agent trained by the generated

data achieved a score faster than pure DDQN. Additionally, RLfGD has a slightly higher score than pure DDQN. The agent achieved 30 points after only 400 iterations. This means that the agent learned to avoid other vehicles and maintain a high speed. In the intersection, differences between the two algorithms are relatively obvious. The vehicle left turn at intersections is characterized by a sparse rewards problem. Thanks to the demonstration data, the agent performed well in the initial training episode and reached a high reward earlier than the DDQN algorithm. The results demonstrate that pre-training reduces the possibility of the agent exploring an unreasonable range of action. Meanwhile, putting the demonstration data into the replay buffer can lead to higher rewards for the agent. As the agent gets a high reward easily via exploration and ER, the reward difference is not significant in the highway environment. However, it is difficult for agents to achieve higher rewards in the intersection environment and the disparity in rewards is more pronounced.

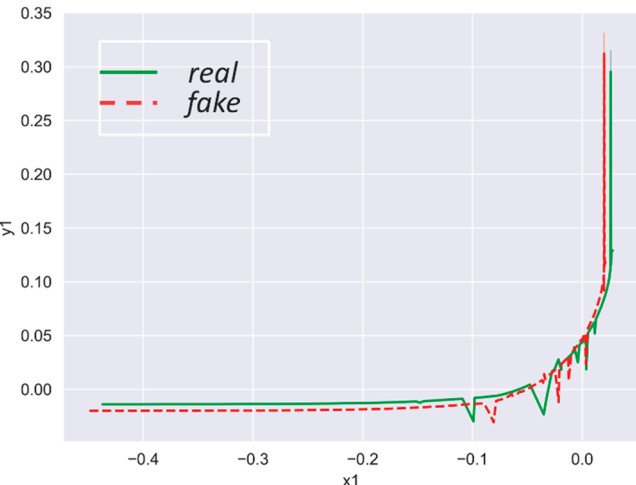

**Figure 7.** The real and generated trajectories of ego-vehicles in intersection environments; (x1, y1) are the coordinates of the ego-vehicle.

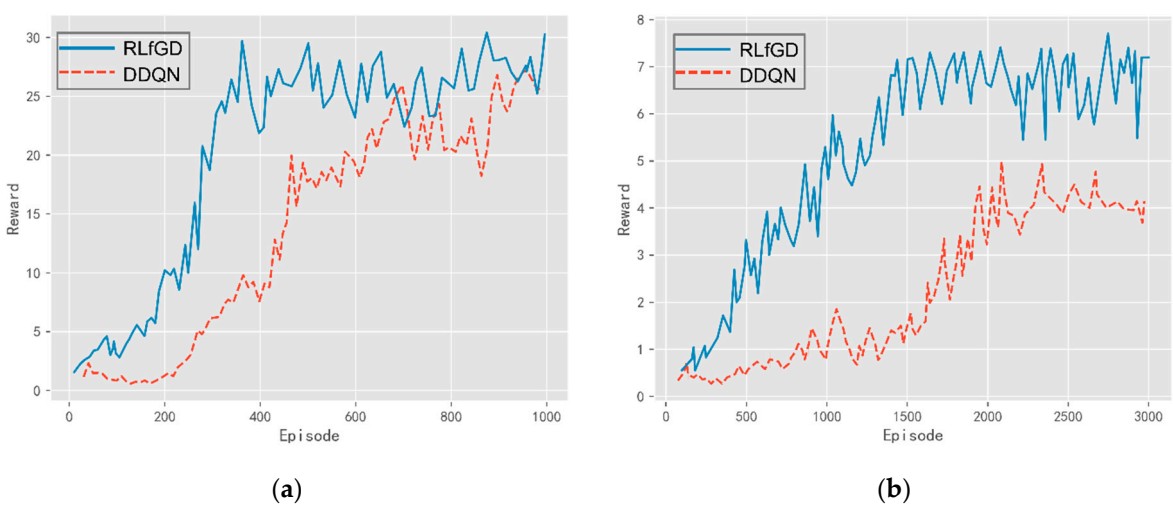

(**a**)                                              (**b**)

**Figure 8.** The comparison results of the average episode rewards. (**a**): The performance of different algorithms on the highway. (**b**): The performance of different algorithms in the intersection environment.

## 5. Conclusions

In this paper, a method of reinforcement learning from generated data (RLfGD) is presented, which consists of a generative model and a learning model. With the generative model, we solved the problem of collecting demonstration data on sparse tasks. The experimental results show that the generative data are close to the real data. In addition,

we trained classification networks to address the inability to generate discrete action information in demonstration data. We also solved the problem of human demonstration data not all satisfying MDP in the autonomous vehicles' decision-making scenarios. The learning model is based on DQfD; we placed the high-value demonstration data and the experience of the agent interacting with the environment in the DQN experience buffer and sampled it using a high-value priority replay method. This algorithm was deployed in two traffic scenarios. The experiment results demonstrate that in a complex traffic scenario, demonstration data can lead to higher rewards for the agent. The agent reduces the possibility of exploring an unreasonable range of actions after pre-training on demonstration data, so the agent gets a reward when it starts interacting with the environment.

RLfGD provides a novel solution to the sparse reward problem; it addresses the difficulty of collecting demonstration data in sparse reward problems and human demonstration data not fully complying with MDP. Future work will focus on the interpretability of the presentation of the data generation process, allowing for the generation of scenario-specific demonstration data.

**Author Contributions:** Conceptualization, Y.W. and W.X.; methodology, Y.W.; software, S.Q.; validation, Y.W. and X.W.; formal analysis, Y.W.; investigation, Y.W.; resources, W.X.; data curation, Y.W.; writing—original draft preparation, Y.W.; writing—review and editing, Y.W.; visualization, S.Q.; supervision, X.W.; project administration, X.W.; and funding acquisition, X.W. All authors have read and agreed to the published version of the manuscript.

**Funding:** This research received no external funding.

**Institutional Review Board Statement:** "Not applicable" for studies not involving humans or animals.

**Informed Consent Statement:** Informed consent was obtained from all subjects involved in the study.

**Data Availability Statement:** Not applicable.

**Acknowledgments:** The authors would like to express their appreciation to the developers of Pytorch and deep Q-networks.

**Conflicts of Interest:** The authors declare no conflict of interest.

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
