# Peer review of "A Method for High-Value Driving Demonstration Data Generation Based on One-Dimensional Deep Convolutional Generative Adversarial Networks"

_electronics, doi:10.3390/electronics11213553_

Round 1

Reviewer 1 Report

Various parts of the manuscript needs to be improved:

- Use of the English language in the manuscript should be revised (academic writing style). Personal pronouns, such as "we" or "our", should not be used in the manuscript. In the manuscript neutral and impersonal language should be used.

- Introduction / Literature review part of the manuscript is too limited. Proper and detailed literature review should be performed, i.e. more research papers should be analysed in more detailed way. This is a major drawback of the manuscript.

- Another drawback of the introduction – research problem and aim of the work should be named very clearly. In general, development of new model is not a research problem. Thus, mind-flow needs to be improved.

- Section 2 – Background of what? Title of the section is misleading. If section 2 describes previously developed framework, why references are not provided?

- Another major drawback of the manuscript – section 3. The description of the approach development is very limited. Everything should be explained in significantly more detailed way: from the assumptions and equations, till the entire framework.

- Analysis of the result also should be significantly improved. As it is now, analysis is written in descriptive style. There is no actual proper critical analysis of the results. 

- Conclusions do not properly represents performed research.

Author Response

Dear Reviewers, first of all, we would like to thank you for your comments on our manuscript, we have revised the manuscript according to your comments, and the specific revisions are shown in the cover letter.

Reviewer 2 Report

In this paper, the authors propose a reinforcement learning from generated data (RLfGD) method that consists of a generative model and a learning model. The 1-DGAN generative model introduces a method for generating high-value demonstration data. To generate one-dimensional data, both the generator and the discriminator are built from one-dimensional convolutional neural networks. Furthermore, classification networks are trained to address the inability of demonstration data to generate discrete action information. The DQfD-based learning model, which places high-value demonstration data in the DQN experience buffer and samples it using a high-value priority replay method, greatly improves the DQN's efficiency. This algorithm is used in two different traffic scenarios.    The experiments show that generating data can speed up the process of reinforcement learning. Performance enhancement is obvious, especially in complex scenarios. The sparse reward problem is addressed by RLfGD, which addresses the difficulty of collecting demonstration data in sparse reward problems as well as human demonstration data that is not fully compliant with MDP.   All in all, this is a good paper with good novelty and contributions. I recommend accepting it. The authors may like to improve the textual and graphical presentation of the paper.

Author Response

(The authors gave the same response as above.)

Reviewer 3 Report

This paper tries to use a generative neural network to compensate for the lack of experience in imitation reinforcement learning. This idea is interesting but the work needs polishing. I have concerns regarding the learning process of the algorithms. Detailed comments are shown below:

What are the input and output of the generative neural network?

1. If the algorithm only focuses on the high-reward transitions but does not have enough real data, it could easily fall into the local optimum due to a lack of exploration. How would the authors address that?

2. How is the comparison DDQN algorithm being trained? Are fake data used?

3. If human behavior does not satisfy MDP, how would Q learning work? The whole proof and convergence analysis of Q learning is built on the assumption that the physical quantities can be formulated into an MDP. Is there theoretical proof of non-MDP Q learning?

Author Response

(The authors gave the same response as above.)

Reviewer 4 Report

1.The manuscript presents a method of reinforcement learning from generated data (RLfGD), which consists of a generative model and a learning model. The generative model is to generate high-value demonstration data.  Both the models are constructed from one-dimensional convolutional neural networks to produce one-dimensional data. Th idea of the work is to provides a novel solution to the sparse reward problem that is to overcome the difficulty of collecting demonstration data in sparse reward problems.

2. The paper sounds scientific and could be acceptable for publication with Electronics. 

Author Response

(The authors gave the same response as above.)

Round 2

Reviewer 1 Report

Some parts of the paper still needs to be reconsidered:

- Entire "Literature review" section is still too limited. As it is now, literature review only describes 5 references. That is not enough for proper detailed literature review. As noted previously, more research papers should be analysed in more detailed way. This is still a major drawback of the manuscript.

- How exactly the description of the model development was improved?

- Similarly, how exactly the analysis part was improved? If only couple of sentences were added, I do not see how this is a proper and detailed improvement of the analysis part.

Author Response

Dear Reviewers.
     We have revised the manuscript again according to your request. The details can be found in the pdf file.

Reviewer 3 Report

The authors responded to all my concerns. This manuscript can be accepted with minor proofreading.

Author Response

(The authors gave the same response as above.)

Round 3

Reviewer 1 Report

No new comments.